# Convergence rates of sub-sampled Newton methods

**Murat A. Erdogdu**
Department of Statistics
Stanford University
erdogdu@stanford.edu

**Andrea Montanari**
Department of Statistics
and Electrical Engineering
Stanford University
montanari@stanford.edu

## Abstract

We consider the problem of minimizing a sum of $n$ functions via projected iterations onto a convex parameter set $\mathcal{C} \subset \mathbb{R}^p$, where $n \gg p \gg 1$. In this regime, algorithms which utilize sub-sampling techniques are known to be effective. In this paper, we use sub-sampling techniques together with low-rank approximation to design a new randomized batch algorithm which possesses comparable convergence rate to Newton's method, yet has much smaller per-iteration cost. The proposed algorithm is robust in terms of starting point and step size, and enjoys a composite convergence rate, namely, quadratic convergence at start and linear convergence when the iterate is close to the minimizer. We develop its theoretical analysis which also allows us to select near-optimal algorithm parameters. Our theoretical results can be used to obtain convergence rates of previously proposed sub-sampling based algorithms as well. We demonstrate how our results apply to well-known machine learning problems. Lastly, we evaluate the performance of our algorithm on several datasets under various scenarios.

## 1  Introduction

We focus on the following minimization problem,

$$\text{minimize } f(\theta) \coloneqq \frac{1}{n} \sum_{i=1}^{n} f_i(\theta), \tag{1.1}$$

where $f_i : \mathbb{R}^p \to \mathbb{R}$. Most machine learning models can be expressed as above, where each function $f_i$ corresponds to an observation. Examples include logistic regression, support vector machines, neural networks and graphical models.

Many optimization algorithms have been developed to solve the above minimization problem [Bis95, BV04, Nes04]. For a given convex set $\mathcal{C} \subset \mathbb{R}^p$, we denote the Euclidean projection onto this set by $\mathcal{P}_{\mathcal{C}}$. We consider the updates of the form

$$\hat{\theta}^{t+1} = \mathcal{P}_{\mathcal{C}} \left( \hat{\theta}^t - \eta_t \mathbf{Q}^t \nabla_\theta f(\hat{\theta}^t) \right), \tag{1.2}$$

where $\eta_t$ is the step size and $\mathbf{Q}^t$ is a suitable scaling matrix that provides curvature information. Updates of the form Eq. (1.2) have been extensively studied in the optimization literature (for simplicity, we assume $\mathcal{C} = \mathbb{R}^p$ throughout the introduction). The case where $\mathbf{Q}^t$ is equal to identity matrix corresponds to *Gradient Descent* (GD) which, under smoothness assumptions, achieves linear convergence rate with $\mathcal{O}(np)$ per-iteration cost. More precisely, GD with ideal step size yields $\|\hat{\theta}^{t+1} - \theta_*\|_2 \leq \xi_{1,\text{GD}}^t \|\hat{\theta}^t - \theta_*\|_2$, where, as $\lim_{t\to\infty} \xi_{1,\text{GD}}^t = 1 - (\lambda_p^*/\lambda_1^*)$, and $\lambda_i^*$ is the $i$-th largest eigenvalue of the Hessian of $f(\theta)$ at minimizer $\theta_*$.

Second order methods such as *Newton's Method* (NM) and *Natural Gradient Descent* (NGD) [Ama98] can be recovered by taking $\mathbf{Q}^t$ to be the inverse Hessian and the Fisher information evaluated at the current iterate, respectively. Such methods may achieve quadratic convergence rates with

$\mathcal{O}(np^2 + p^3)$ per-iteration cost [Bis95, Nes04]. In particular, for $t$ large enough, Newton's method yields $\|\hat{\theta}^{t+1} - \theta_*\|_2 \leq \xi_{2,\text{NM}}\|\hat{\theta}^t - \theta_*\|_2^2$, and it is insensitive to the condition number of the Hessian. However, when the number of samples grows large, computing $\mathbf{Q}^t$ becomes extremely expensive.

A popular line of research tries to construct the matrix $\mathbf{Q}^t$ in a way that the update is computationally feasible, yet still provides sufficient second order information. Such attempts resulted in Quasi-Newton methods, in which only gradients and iterates are utilized, resulting in an efficient update on $\mathbf{Q}^t$. A celebrated Quasi-Newton method is the *Broyden-Fletcher-Goldfarb-Shanno* (BFGS) algorithm which requires $\mathcal{O}(np + p^2)$ per-iteration cost [Bis95, Nes04].

An alternative approach is to use *sub-sampling* techniques, where scaling matrix $\mathbf{Q}^t$ is based on randomly selected set of data points [Mar10, BCNN11, VP12, Erd15]. Sub-sampling is widely used in the first order methods, but is not as well studied for approximating the scaling matrix. In particular, theoretical guarantees are still missing.

A key challenge is that the sub-sampled Hessian is close to the actual Hessian along the directions corresponding to large eigenvalues (large curvature directions in $f(\theta)$), but is a poor approximation in the directions corresponding to small eigenvalues (flatter directions in $f(\theta)$). In order to overcome this problem, we use low-rank approximation. More precisely, we treat all the eigenvalues below the $r$-th as if they were equal to the $(r+1)$-th. This yields the desired stability with respect to the sub-sample: we call our algorithm NewSamp. In this paper, we establish the following:

1. NewSamp has a composite convergence rate: quadratic at start and linear near the minimizer, as illustrated in Figure 1. Formally, we prove a bound of the form $\|\hat{\theta}^{t+1} - \theta_*\|_2 \leq \xi_1^t\|\hat{\theta}^t - \theta_*\|_2 + \xi_2^t\|\hat{\theta}^t - \theta_*\|_2^2$ with coefficient that are explicitly given (and are computable from data).
2. The asymptiotic behavior of the linear convergence coefficient is $\lim_{t\to\infty} \xi_1^t = 1 - (\lambda_p^*/\lambda_{r+1}^*) + \delta$, for $\delta$ small. The condition number $(\lambda_1^*/\lambda_p^*)$ which controls the convergence of GD, has been replaced by the milder $(\lambda_{r+1}^*/\lambda_p^*)$. For datasets with strong spectral features, this can be a large improvement, as shown in Figure 1.
3. The above results are achived without tuning the step-size, in particular, by setting $\eta_t = 1$.
4. The complexity per iteration of NewSamp is $\mathcal{O}(np + |S|p^2)$ with $|S|$ the sample size.
5. Our theoretical results can be used to obtain convergence rates of previously proposed sub-sampling algorithms.

The rest of the paper is organized as follows: Section 1.1 surveys the related work. In Section 2, we describe the proposed algorithm and provide the intuition behind it. Next, we present our theoretical results in Section 3, i.e., convergence rates corresponding to different sub-sampling schemes, followed by a discussion on how to choose the algorithm parameters. Two applications of the algorithm are discussed in Section 4. We compare our algorithm with several existing methods on various datasets in Section 5. Finally, in Section 6, we conclude with a brief discussion.

## 1.1 Related Work

Even a synthetic review of optimization algorithms for large-scale machine learning would go beyond the page limits of this paper. Here, we emphasize that the method of choice depends crucially on the amount of data to be used, and their dimensionality (i.e., respectively, on the parameters $n$ and $p$). In this paper, we focus on a regime in which $n$ and $p$ are large but not so large as to make gradient computations (of order $np$) and matrix manipulations (of order $p^3$) prohibitive.

Online algorithms are the option of choice for very large $n$ since the computation per update is independent of $n$. In the case of *Stochastic Gradient Descent* (SGD), the descent direction is formed by a randomly selected gradient. Improvements to SGD have been developed by incorporating the previous gradient directions in the current update equation [SRB13, Bot10, DHS11].

Batch algorithms, on the other hand, can achieve faster convergence and exploit second order information. They are competitive for intermediate $n$. Several methods in this category aim at quadratic, or at least super-linear convergence rates. In particular, Quasi-Newton methods have proven effective [Bis95, Nes04]. Another approach towards the same goal is to utilize sub-sampling to form an approximate Hessian [Mar10, BCNN11, VP12, Erd15]. If the sub-sampled Hessian is close to the true Hessian, these methods can approach NM in terms of convergence rate, nevertheless, they enjoy

---
**Algorithm 1** NewSamp
---

**Input:** $\hat{\theta}^0, r, \epsilon, \{\eta_t\}_t, t = 0$.
     1. **Define:** $\mathcal{P}_\mathcal{C}(\theta) = \operatorname{argmin}_{\theta' \in \mathcal{C}} \|\theta - \theta'\|_2$ is the Euclidean projection onto $\mathcal{C}$,
          $[\mathbf{U}_k, \mathbf{\Lambda}_k] = \text{TruncatedSVD}_k(\mathbf{H})$ is rank-$k$ truncated SVD of $\mathbf{H}$ with $\mathbf{\Lambda}_{ii} = \lambda_i$.
     2. **while** $\|\hat{\theta}^{t+1} - \hat{\theta}^t\|_2 \leq \epsilon$ **do**
        Sub-sample a set of indices $S_t \subset [n]$.
        Let $\mathbf{H}_{S_t} = \frac{1}{|S_t|} \sum_{i \in S_t} \mathbf{\nabla}_\theta^2 f_i(\hat{\theta}^t)$, and $[\mathbf{U}_{r+1}, \mathbf{\Lambda}_{r+1}] = \text{TruncatedSVD}_{r+1}(\mathbf{H}_{S_t})$,
        $\mathbf{Q}^t = \lambda_{r+1}^{-1} \mathbf{I}_p + \mathbf{U}_r \left( \mathbf{\Lambda}_r^{-1} - \lambda_{r+1}^{-1} \mathbf{I}_r \right) \mathbf{U}_r^T$,
        $\hat{\theta}^{t+1} = \mathcal{P}_\mathcal{C} \left( \hat{\theta}^t - \eta_t \mathbf{Q}^t \nabla_\theta f(\hat{\theta}^t) \right)$,
        $t \leftarrow t + 1$.
     3. **end while**
**Output:** $\hat{\theta}^t$.

---

much smaller complexity per update. No convergence rate analysis is available for these methods: this analysis is the main contribution of our paper. To the best of our knowledge, the best result in this direction is proven in [BCNN11] that estabilishes asymptotic convergence without quantitative bounds (exploiting general theory from [GNS09]).

On the further improvements of the sub-sampling algorithms, a common approach is to use *Conjugate Gradient* (CG) methods and/or Krylov sub-spaces [Mar10, BCNN11, VP12]. Lastly, there are various hybrid algorithms that combine two or more techniques to increase the performance. Examples include, sub-sampling and Quasi-Newton [BHNS14], SGD and GD [FS12], NGD and NM [LRF10], NGD and low-rank approximation [LRMB08].

## 2 NewSamp : Newton-Sampling method via rank thresholding

In the regime we consider, $n \gg p$, there are two main drawbacks associated with the classical second order methods such as Newton's method. The dominant issue is the computation of the Hessian matrix, which requires $\mathcal{O}(np^2)$ operations, and the other issue is inverting the Hessian, which requires $\mathcal{O}(p^3)$ computation. Sub-sampling is an effective and efficient way of tackling the first issue. Recent empirical studies show that sub-sampling the Hessian provides significant improvement in terms of computational cost, yet preserves the fast convergence rate of second order methods [Mar10, VP12]. If a uniform sub-sample is used, the sub-sampled Hessian will be a random matrix with expected value at the true Hessian, which can be considered as a sample estimator to the mean. Recent advances in statistics have shown that the performance of various estimators can be significantly improved by simple procedures such as *shrinkage* and/or *thresholding* [CCS10, DGJ13]. To this extent, we use low-rank approximation as the important second order information is generally contained in the largest few eigenvalues/vectors of the Hessian.

NewSamp is presented as Algorithm 1. At iteration step $t$, the sub-sampled set of indices, its size and the corresponding sub-sampled Hessian is denoted by $S_t$, $|S_t|$ and $\mathbf{H}_{S_t}$, respectively. Assuming that the functions $f_i$'s are convex, eigenvalues of the symmetric matrix $\mathbf{H}_{S_t}$ are non-negative. Therefore, SVD and eigenvalue decomposition coincide. The operation $\text{TruncatedSVD}_k(\mathbf{H}_{S_t}) = [\mathbf{U}_k, \mathbf{\Lambda}_k]$ is the best rank-$k$ approximation, i.e., takes $\mathbf{H}_{S_t}$ as input and returns the largest $k$ eigenvalues $\mathbf{\Lambda}_k \in \mathbb{R}^{k \times k}$ with the corresponding $k$ eigenvectors $\mathbf{U}_k \in \mathbb{R}^{p \times k}$. This procedure requires $\mathcal{O}(kp^2)$ computation [HMT11]. Operator $\mathcal{P}_\mathcal{C}$ projects the current iterate to the feasible set $\mathcal{C}$ using Euclidean projection. We assume that this projection can be done efficiently. To construct the curvature matrix $[\mathbf{Q}^t]^{-1}$, instead of using the basic rank-$r$ approximation, we fill its 0 eigenvalues with the $(r+1)$-th eigenvalue of the sub-sampled Hessian which is the largest eigenvalue below the threshold. If we compute a truncated SVD with $k = r+1$ and $\mathbf{\Lambda}_{ii} = \lambda_i$, the described operation results in

$$\mathbf{Q}^t = \lambda_{r+1}^{-1} \mathbf{I}_p + \mathbf{U}_r \left( \mathbf{\Lambda}_r^{-1} - \lambda_{r+1}^{-1} \mathbf{I}_r \right) \mathbf{U}_r^T, \tag{2.1}$$

which is simply the sum of a scaled identity matrix and a rank-$r$ matrix. Note that the low-rank approximation that is suggested to improve the curvature estimation has been further utilized to reduce the cost of computing the inverse matrix. Final per-iteration cost of NewSamp will be $\mathcal{O}\left(np + (|S_t| + r)p^2\right) \approx \mathcal{O}\left(np + |S_t|p^2\right)$. NewSamp takes the parameters $\{\eta_t, |S_t|\}_t$ and $r$ as inputs. We discuss in Section 3.4, how to choose them optimally, based on the theory in Section 3.

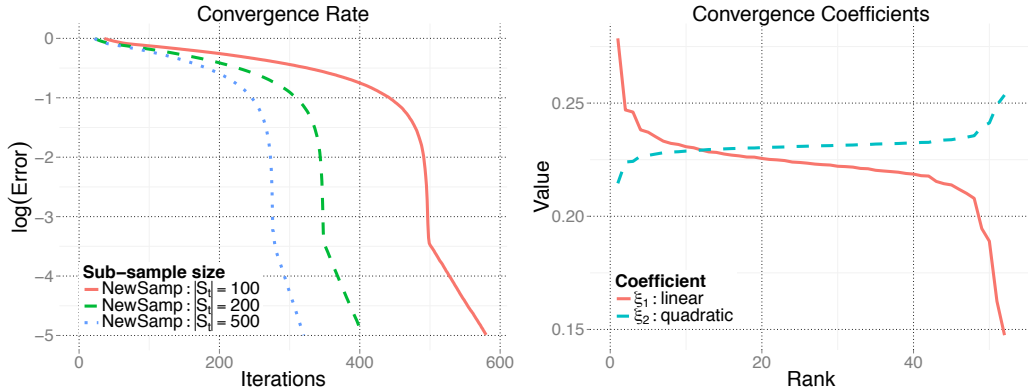

Figure 1: Left plot demonstrates convergence rate of NewSamp , which starts with a quadratic rate and transitions into linear convergence near the true minimizer. The right plot shows the effect of eigenvalue thresholding on the convergence coefficients up to a scaling constant. $x$-axis shows the number of kept eigenvalues. Plots are obtained using *Covertype* dataset.

By the construction of $\mathbf{Q}^t$, NewSamp will always be a descent algorithm. It enjoys a quadratic convergence rate at start which transitions into a linear rate in the neighborhood of the minimizer. This behavior can be observed in Figure 1. The left plot in Figure 1 shows the convergence behavior of NewSamp over different sub-sample sizes. We observe that large sub-samples result in better convergence rates as expected. As the sub-sample size increases, slope of the linear phase decreases, getting closer to that of quadratic phase. We will explain this phenomenon in Section 3, by Theorems 3.2 and 3.3. The right plot in Figure 1 demonstrates how the coefficients of two phases depend on the thresholded rank. Coefficient of the quadratic phase increases with the rank threshold, whereas for the linear phase, relation is reversed.

## 3 Theoretical results

In this section, we provide the convergence analysis of NewSamp based on two different sub-sampling schemes:

- S1: **Independent sub-sampling**: At each iteration $t$, $S_t$ is uniformly sampled from $[n] = \{1, 2, ..., n\}$, independently from the sets $\{S_\tau\}_{\tau < t}$, with or without replacement.
- S2: **Sequentially dependent sub-sampling**: At each iteration $t$, $S_t$ is sampled from $[n]$, based on a distribution which might depend on the previous sets $\{S_\tau\}_{\tau < t}$, but not on any randomness in the data.

The first sub-sampling scheme is simple and commonly used in optimization. One drawback is that the sub-sampled set at the current iteration is independent of the previous sub-samples, hence does not consider which of the samples were previously used to form the approximate curvature information. In order to prevent cycles and obtain better performance near the optimum, one might want to increase the sample size as the iteration advances [Mar10], including previously unused samples. This process results in a sequence of dependent sub-samples which falls into the sub-sampling scheme S2. In our theoretical analysis, we make the following assumptions:

**Assumption 1** (Lipschitz continuity). *For any subset $S \subset [n]$, $\exists M_{|S|}$ depending on the size of $S$, such that $\forall \theta, \theta' \in \mathcal{C}$,*
$$\|\mathbf{H}_S(\theta) - \mathbf{H}_S(\theta')\|_2 \leq M_{|S|} \|\theta - \theta'\|_2.$$

**Assumption 2** (Bounded Hessian). *$\forall i \in [n]$, $\boldsymbol{\nabla}_\theta^2 f_i(\theta)$ is upper bounded by a constant $K$, i.e.,*
$$\max_{i \leq n} \left\|\boldsymbol{\nabla}_\theta^2 f_i(\theta)\right\|_2 \leq K.$$

### 3.1 Independent sub-sampling

In this section, we assume that $S_t \subset [n]$ is sampled according to the sub-sampling scheme S1. In fact, many stochastic algorithms assume that $S_t$ is a uniform subset of $[n]$, because in this case the sub-sampled Hessian is an unbiased estimator of the full Hessian. That is, $\forall \theta \in \mathcal{C}$, $\mathbb{E}\left[\mathbf{H}_{S_t}(\theta)\right] = \mathbf{H}_{[n]}(\theta)$, where the expectation is over the randomness in $S_t$. We next show that for any scaling matrix $\mathbf{Q}^t$ that is formed by the sub-samples $S_t$, iterations of the form Eq. (1.2) will have a composite convergence rate, i.e., combination of a linear and a quadratic phases.

**Lemma 3.1.** *Assume that the parameter set $\mathcal{C}$ is convex and $S_t \subset [n]$ is based on sub-sampling scheme S1 and sufficiently large. Further, let the Assumptions 1 and 2 hold and $\theta_* \in \mathcal{C}$. Then, for an absolute constant $c > 0$, with probability at least $1 - 2/p$, the updates of the form Eq. (1.2) satisfy*

$$\|\hat{\theta}^{t+1} - \theta_*\|_2 \leq \xi_1^t \|\hat{\theta}^t - \theta_*\|_2 + \xi_2^t \|\hat{\theta}^t - \theta_*\|_2^2,$$

*for coefficients $\xi_1^t$ and $\xi_2^t$ defined as*

$$\xi_1^t = \left\| I - \eta_t \mathbf{Q}^t \mathbf{H}_{S_t}(\hat{\theta}^t) \right\|_2 + \eta_t cK \left\| \mathbf{Q}^t \right\|_2 \sqrt{\frac{\log(p)}{|S_t|}}, \qquad \xi_2^t = \eta_t \frac{M_n}{2} \left\| \mathbf{Q}^t \right\|_2.$$

**Remark 1.** *If the initial point $\hat{\theta}^0$ is close to $\theta_*$, the algorithm will start with a quadratic rate of convergence which will transform into linear rate later in the close neighborhood of the optimum.*

The above lemma holds for any matrix $\mathbf{Q}^t$. In particular, if we choose $\mathbf{Q}^t = \mathbf{H}_{S_t}^{-1}$, we obtain a bound for the simple sub-sampled Hessian method. In this case, the coefficients $\xi_1^t$ and $\xi_2^t$ depend on $\|\mathbf{Q}^t\|_2 = 1/\lambda_p^t$ where $\lambda_p^t$ is the smallest eigenvalue of the sub-sampled Hessian. Note that $\lambda_p^t$ can be arbitrarily small which might blow up both of the coefficients. In the following, we will see how NewSamp remedies this issue.

**Theorem 3.2.** *Let the assumptions in Lemma 3.1 hold. Denote by $\lambda_i^t$, the $i$-th eigenvalue of $\mathbf{H}_{S_t}(\hat{\theta}^t)$ where $\hat{\theta}^t$ is given by NewSamp at iteration step $t$. If the step size satisfies*

$$\eta_t \leq \frac{2}{1 + \lambda_p^t/\lambda_{r+1}^t}, \tag{3.1}$$

*then we have, with probability at least $1 - 2/p$,*

$$\|\hat{\theta}^{t+1} - \theta_*\|_2 \leq \xi_1^t \|\hat{\theta}^t - \theta_*\|_2 + \xi_2^t \|\hat{\theta}^t - \theta_*\|_2^2,$$

*for an absolute constant $c > 0$, for the coefficients $\xi_1^t$ and $\xi_2^t$ are defined as*

$$\xi_1^t = 1 - \eta_t \frac{\lambda_p^t}{\lambda_{r+1}^t} + \eta_t \frac{cK}{\lambda_{r+1}^t} \sqrt{\frac{\log(p)}{|S_t|}}, \qquad \xi_2^t = \eta_t \frac{M_n}{2\lambda_{r+1}^t}.$$

NewSamp has a composite convergence rate where $\xi_1^t$ and $\xi_2^t$ are the coefficients of the linear and the quadratic terms, respectively (See the right plot in Figure 1). We observe that the sub-sampling size has a significant effect on the linear term, whereas the quadratic term is governed by the Lipschitz constant. We emphasize that the case $\eta_t = 1$ is feasible for the conditions of Theorem 3.2.

## 3.2 Sequentially dependent sub-sampling

Here, we assume that the sub-sampling scheme S2 is used to generate $\{S_\tau\}_{\tau \geq 1}$. Distribution of sub-sampled sets may depend on each other, but not on any randomness in the dataset. Examples include fixed sub-samples as well as sub-samples of increasing size, sequentially covering unused data. In addition to Assumptions 1-2, we assume the following.

**Assumption 3** (i.i.d. observations). *Let $z_1, z_2, ..., z_n \in Z$ be i.i.d. observations from a distribution $\mathcal{D}$. For a fixed $\theta \in \mathbb{R}^p$ and $\forall i \in [n]$, we assume that the functions $\{f_i\}_{i=1}^n$ satisfy $f_i(\theta) = \varphi(z_i, \theta)$, for some function $\varphi : Z \times \mathbb{R}^p \to \mathbb{R}$.*

Most statistical learning algorithms can be formulated as above, e.g., in classification problems, one has access to i.i.d. samples $\{(y_i, x_i)\}_{i=1}^n$ where $y_i$ and $x_i$ denote the class label and the covariate, and $\varphi$ measures the classification error (See Section 4 for examples). For sub-sampling scheme S2, an analogue of Lemma 3.1 is stated in Appendix as Lemma B.1, which leads to the following result.

**Theorem 3.3.** *Assume that the parameter set $\mathcal{C}$ is convex and $S_t \subset [n]$ is based on the sub-sampling scheme S2. Further, let the Assumptions 1, 2 and 3 hold, almost surely. Conditioned on the event $\mathcal{E} = \{\theta_* \in \mathcal{C}\}$, if the step size satisfies Eq. 3.1, then for $\hat{\theta}^t$ given by NewSamp at iteration $t$, with probability at least $1 - c_{\mathcal{E}} e^{-p}$ for $c_{\mathcal{E}} = c/\mathbb{P}(\mathcal{E})$, we have*

$$\|\hat{\theta}^{t+1} - \theta_*\|_2 \leq \xi_1^t \|\hat{\theta}^t - \theta_*\|_2 + \xi_2^t \|\hat{\theta}^t - \theta_*\|_2^2,$$

*for the coefficients $\xi_1^t$ and $\xi_2^t$ defined as*

$$\xi_1^t = 1 - \eta_t \frac{\lambda_p^t}{\lambda_{r+1}^t} + \eta_t \frac{c'K}{\lambda_{r+1}^t} \sqrt{\frac{p}{|S_t|} \log \left( \frac{\text{diam}(\mathcal{C})^2 \left( M_n + M_{|S_t|} \right)^2 |S_t|}{K^2} \right)}, \qquad \xi_2^t = \eta_t \frac{M_n}{2\lambda_{r+1}^t},$$

*where $c, c' > 0$ are absolute constants and $\lambda_i^t$ denotes the $i$-th eigenvalue of $\mathbf{H}_{S_t}(\hat{\theta}^t)$.*

Compared to the Theorem 3.2, we observe that the coefficient of the quadratic term does not change. This is due to Assumption 1. However, the bound on the linear term is worse, since we use the uniform bound over the convex parameter set $\mathcal{C}$.

## 3.3 Dependence of coefficients on $t$ and convergence guarantees

The coefficients $\xi_1^t$ and $\xi_2^t$ depend on the iteration step which is an undesirable aspect of the above results. However, these constants can be well approximated by their analogues $\xi_1^*$ and $\xi_2^*$ evaluated at the optimum which are defined by simply replacing $\lambda_j^t$ with $\lambda_j^*$ in their definition, where the latter is the $j$-th eigenvalue of full-Hessian at $\theta_*$. For the sake of simplicity, we only consider the case where the functions $\theta \to f_i(\theta)$ are quadratic.

**Theorem 3.4.** *Assume that the functions $f_i(\theta)$ are quadratic, $S_t$ is based on scheme S1 and $\eta_t = 1$. Let the full Hessian at $\theta_*$ be lower bounded by $k$. Then for sufficiently large $|S_t|$ and absolute constants $c_1, c_2$, with probability $1 - 2/p$*

$$\left| \xi_1^t - \xi_1^* \right| \leq \frac{c_1 K \sqrt{\log(p)/|S_t|}}{k \left( k - c_2 K \sqrt{\log(p)/|S_t|} \right)} := \delta.$$

Theorem 3.4 implies that, when the sub-sampling size is sufficiently large, $\xi_1^t$ will concentrate around $\xi_1^*$. Generalizing the above theorem to non-quadratic functions is straightforward, in which case, one would get additional terms involving the difference $\|\hat{\theta}^t - \theta_*\|_2$. In the case of scheme S2, if one uses fixed sub-samples, then the coefficient $\xi_1^t$ does not depend on $t$. The following corollary gives a sufficient condition for convergence. A detailed discussion on the number of iterations until convergence and further local convergence properties can be found in [Erd15, EM15].

**Corollary 3.5.** *Assume that $\xi_1^t$ and $\xi_2^t$ are well-approximated by $\xi_1^*$ and $\xi_2^*$ with an error bound of $\delta$, i.e., $\xi_i^t \leq \xi_i^* + \delta$ for $i = 1, 2$, as in Theorem 3.4. For the initial point $\hat{\theta}^0$, a sufficient condition for convergence is*

$$\|\hat{\theta}^0 - \theta_*\|_2 < \frac{1 - \xi_1^* - \delta}{\xi_2^* + \delta}.$$

## 3.4 Choosing the algorithm parameters

*Step size:* Let $\gamma = \mathcal{O}(\log(p)/|S_t|)$. We suggest the following step size for NewSamp at iteration $t$,

$$\eta_t(\gamma) = \frac{2}{1 + \lambda_p^t / \lambda_{r+1}^t + \gamma}. \tag{3.2}$$

Note that $\eta_t(0)$ is the upper bound in Theorems 3.2 and 3.3 and it minimizes the first component of $\xi_1^t$. The other terms in $\xi_1^t$ and $\xi_2^t$ linearly depend on $\eta_t$. To compensate for that, we shrink $\eta_t(0)$ towards 1. Contrary to most algorithms, optimal step size of NewSamp is larger than 1. A rigorous derivation of Eq. 3.2 can be found in [EM15].

*Sample size:* By Theorem 3.2, a sub-sample of size $\mathcal{O}((K/\lambda_p^*)^2 \log(p))$ should be sufficient to obtain a small coefficient for the linear phase. Also note that sub-sample size $|S_t|$ scales quadratically with the condition number.

*Rank threshold:* For a full-Hessian with effective rank $R$ (trace divided by the largest eigenvalue), it suffices to use $\mathcal{O}(R \log(p))$ samples [Ver10]. Effective rank is upper bounded by the dimension $p$. Hence, one can use $p \log(p)$ samples to approximate the full-Hessian and choose a rank threshold which retains the important curvature information.

# 4 Examples

## 4.1 Generalized Linear Models (GLM)

Maximum likelihood estimation in a GLM setting is equivalent to minimizing the negative log-likelihood $\ell(\theta)$,

$$\underset{\theta \in \mathcal{C}}{\text{minimize}} \; f(\theta) = \frac{1}{n} \sum_{i=1}^{n} \left[ \Phi(\langle x_i, \theta \rangle) - y_i \langle x_i, \theta \rangle \right], \tag{4.1}$$

where $\Phi$ is the *cumulant generating function*, $x_i \in \mathbb{R}^p$ denote the rows of design matrix $\mathbf{X} \in \mathbb{R}^{n \times p}$, and $\theta \in \mathbb{R}^p$ is the coefficient vector. Here, $\langle x, \theta \rangle$ denotes the inner product between the vectors $x$, $\theta$. The function $\Phi$ defines the type of GLM, i.e., $\Phi(z) = z^2$ gives ordinary least squares (OLS) and $\Phi(z) = \log(1 + e^z)$ gives logistic regression (LR). Using the results from Section 3, we perform a convergence analysis of our algorithm on a GLM problem.

**Corollary 4.1.** *Let $S_t \subset [n]$ be a uniform sub-sample, and $\mathcal{C} = \mathbb{R}^p$ be the parameter set. Assume that the second derivative of the cumulant generating function, $\Phi^{(2)}$ is bounded by 1, and it is Lipschitz continuous with Lipschitz constant $L$. Further, assume that the covariates are contained in a ball of radius $\sqrt{R_x}$, i.e. $\max_{i \in [n]} \|x_i\|_2 \leq \sqrt{R_x}$. Then, for $\hat{\theta}^t$ given by NewSamp with constant step size $\eta_t = 1$ at iteration $t$, with probability at least $1 - 2/p$, we have*

$$\|\hat{\theta}^{t+1} - \theta_*\|_2 \leq \xi_1^t \|\hat{\theta}^t - \theta_*\|_2 + \xi_2^t \|\hat{\theta}^t - \theta_*\|_2^2,$$

*for constants $\xi_1^t$ and $\xi_2^t$ defined as*

$$\xi_1^t = 1 - \frac{\lambda_i^t}{\lambda_{r+1}^t} + \frac{cR_x}{\lambda_{r+1}^t} \sqrt{\frac{\log(p)}{|S_t|}}, \qquad\qquad \xi_2^t = \frac{LR_x^{3/2}}{2\lambda_{r+1}^t},$$

*where $c > 0$ is an absolute constant and $\lambda_i^t$ is the ith eigenvalue of $\mathbf{H}_{S_t}(\hat{\theta}^t)$.*

## 4.2 Support Vector Machines (SVM)

A linear SVM provides a *separating hyperplane* which maximizes the *margin*, i.e., the distance between the hyperplane and the support vectors. Although the vast majority of the literature focuses on the dual problem [SS02], SVMs can be trained using the primal as well. Since the dual problem does not scale well with the number of data points (some approaches get $\mathcal{O}(n^3)$ complexity) the primal might be better-suited for optimization of linear SVMs [Cha07]. The primal problem for the linear SVM can be written as

$$\underset{\theta \in \mathcal{C}}{\text{minimize}} \; f(\theta) = \frac{1}{2}\|\theta\|_2^2 + \frac{1}{2}C\sum_{i=1}^{n}\ell(y_i, \langle\theta, x_i\rangle) \tag{4.2}$$

where $(y_i, x_i)$ denote the data samples, $\theta$ defines the separating hyperplane, $C > 0$ and $\ell$ could be any loss function. The most commonly used loss functions include *Hinge-p loss*, *Huber loss* and their smoothed versions [Cha07]. Smoothing or approximating such losses with more stable functions is sometimes crucial in optimization. In the case of NewSamp which requires the loss function to be twice differentiable (almost everywhere), we suggest either smoothed Huber loss, or Hinge-2 loss [Cha07]. In the case of Hinge-2 loss, i.e., $\ell(y, \langle\theta, x\rangle) = \max\{0, 1 - y\langle\theta, x\rangle\}^2$, by combining the offset and the normal vector of the hyperplane into a single parameter vector $\theta$, and denoting by $SV_t$ the set of indices of all the support vectors at iteration $t$, we may write the Hessian,

$$\nabla_\theta^2 f(\theta) = \frac{1}{|SV_t|}\Big\{\mathbf{I} + C\sum_{i \in SV_t}x_i x_i^T\Big\}, \qquad \text{where} \qquad SV_t = \{i : y_i\langle\theta^t, x_i\rangle < 1\}.$$

When $|SV_t|$ is large, the problem falls into our setup and can be solved efficiently using NewSamp. Note that unlike the GLM setting, Lipschitz condition of our Theorems do not apply here. However, we empirically demonstrate that NewSamp works regardless of such assumptions.

## 5 Experiments

In this section, we validate the performance of NewSamp through numerical studies. We experimented on two optimization problems, namely, *Logistic Regression* (LR) and SVM. LR minimizes Eq. 4.1 for the logistic function, whereas SVM minimizes Eq. 4.2 for the Hinge-2 loss. In the following, we briefly describe the algorithms that are used in the experiments:

1. *Gradient Descent* (GD), at each iteration, takes a step proportional to negative of the full gradient evaluated at the current iterate. Under certain regularity conditions, GD exhibits a linear convergence rate.
2. *Accelerated Gradient Descent* (AGD) is proposed by Nesterov [Nes83], which improves over the gradient descent by using a momentum term.
3. *Newton's Method* (NM) achieves a quadratic convergence rate by utilizing the inverse Hessian evaluated at the current iterate.
4. *Broyden-Fletcher-Goldfarb-Shanno* (BFGS) is the most popular and stable Quasi-Newton method. $\mathbf{Q}^t$ is formed by accumulating the information from iterates and gradients.
5. *Limited Memory BFGS* (L-BFGS) is a variant of BFGS, which uses only the recent iterates and gradients to construct $\mathbf{Q}^t$, providing improvement in terms of memory usage.
6. *Stochastic Gradient Descent* (SGD) is a simplified version of GD where, at each iteration, a randomly selected gradient is used. We follow the guidelines of [Bot10] for the step size.

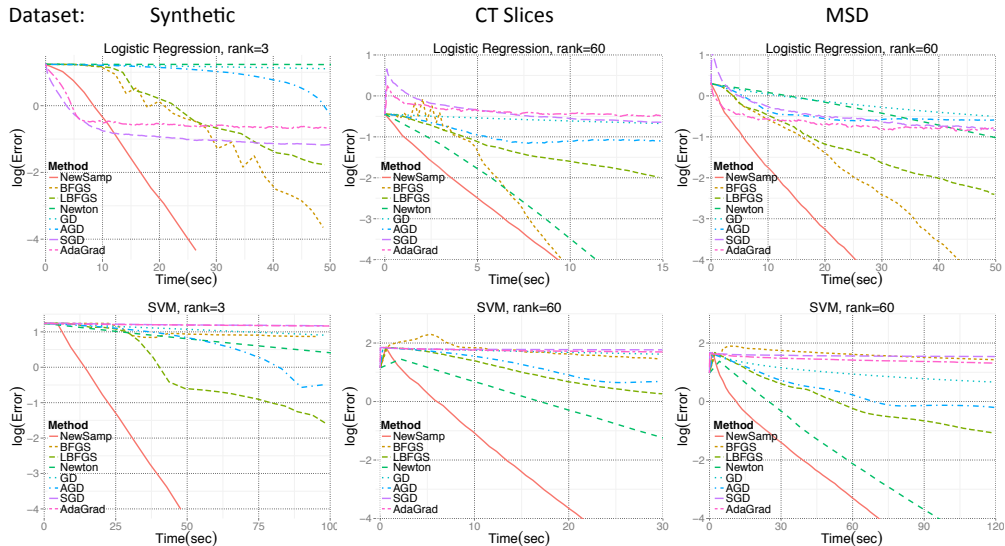

Figure 2: Performance of several algorithms on different datasets. NewSamp is represented with red color .

7. *Adaptive Gradient Scaling* (AdaGrad) uses an adaptive learning rate based on the previous gradients. AdaGrad significantly improves the performance and stability of SGD [DHS11].

For batch algorithms, we used constant step size and for all the algorithms, the step size that provides the fastest convergence is chosen. For stochastic algorithms, we optimized over the parameters that define the step size. Parameters of NewSamp are selected following the guidelines in Section 3.4.

We experimented over various datasets that are given in Table 1. Each dataset consists of a design matrix $\mathbf{X} \in \mathbb{R}^{n \times p}$ and the corresponding observations (classes) $y \in \mathbb{R}^n$. Synthetic data is generated through a multivariate Gaussian distribution. As a methodological choice, we selected moderate values of $p$, for which Newton's method can still be implemented, and nevertheless we can demonstrate an improvement. For larger values of $p$, comparison is even more favorable to our approach.

The effects of sub-sampling size $|S_t|$ and rank threshold are demonstrated in Figure 1. A thorough comparison of the aforementioned optimization techniques is presented in Figure 2. In the case of LR, we observe that stochastic methods enjoy fast convergence at start, but slows down after several epochs. The algorithm that comes close to NewSamp in terms of performance is BFGS. In the case of SVM, NM is the closest algorithm to NewSamp . Note that the global convergence of BFGS is not better than that of GD [Nes04]. The condition for super-linear rate is $\sum_t \|\theta^t - \theta_*\|_2 < \infty$ for which, an initial point close to the optimum is required [DM77]. This condition can be rarely satisfied in practice, which also affects the performance of other second order methods. For NewSamp, even though rank thresholding provides a level of robustness, we found that initial point is still an important factor. Details about Figure 2 and additional experiments can be found in Appendix C.

| Dataset | $n$ | $p$ | $r$ | Reference |
|---------|-----|-----|-----|-----------|
| CT slices | 53500 | 386 | 60 | [GKS$^+$11, Lic13] |
| Covertype | 581012 | 54 | 20 | [BD99, Lic13] |
| MSD | 515345 | 90 | 60 | [MEWL, Lic13] |
| Synthetic | 500000 | 300 | 3 | – |

Table 1: Datasets used in the experiments.

## 6 Conclusion

In this paper, we proposed a sub-sampling based second order method utilizing low-rank Hessian estimation. The proposed method has the target regime $n \gg p$ and has $\mathcal{O}\left(np + |S|p^2\right)$ complexity per-iteration. We showed that the convergence rate of NewSamp is composite for two widely used sub-sampling schemes, i.e., starts as quadratic convergence and transforms to linear convergence near the optimum. Convergence behavior under other sub-sampling schemes is an interesting line of research. Numerical experiments demonstrate the performance of the proposed algorithm which we compared to the classical optimization methods.

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
