[Supplementary Material · appendix.pdf]

# Appendix

# Convergence rates of sub-sampled Newton methods

## A   Quadratic functions

In the case of quadratic functions, since the Lipschitz constant is $0$ , we obtain $\xi_2^t = 0$ and the algorithm converges linearly. Following corollary summarizes this case when the sub-sampling scheme S1 is used.

**Corollary A.1** (Quadratic functions). *Let the assumptions of Theorem 3.2 hold. Assume that $\forall i \in [n]$, the functions $\theta : \mathbb{R}^p \to f_i(\theta)$ are quadratic. Then, for $\hat{\theta}^t$ given by NewSamp at iteration step $t$, with probability at least $1 - 2/p$, for the coefficient $\xi_1^t$ defined as in Theorem 3.2, we have*

$$\|\hat{\theta}^{t+1} - \theta_*\|_2 \leq \xi_1^t \|\hat{\theta}^t - \theta_*\|_2. \tag{A.1}$$

Similar to Corollary A.1, we have the following result for the quadratic functions under sub-sampling scheme S2.

**Corollary A.2** (Quadratic functions). *Let the assumptions of Theorem 3.3 hold. Further assume that $\forall i \in [n]$, the functions $\theta \to f_i(\theta)$ are quadratic. Conditioned on $\mathcal{E}$, with probability at least $1 - c\,e^{-p}$, NewSamp iterates satisfy*

$$\|\hat{\theta}^{t+1} - \theta_*\|_2 \leq \xi_1^t \|\hat{\theta}^t - \theta_*\|_2,$$

*for coefficient $\xi_1^t$ defined as in Theorem 3.3.*

## B   Proofs of Theorems and Lemmas

*Proof of Lemma 3.1.* We write

$$\hat{\theta}^t - \theta_* - \eta_t \mathbf{Q}^t \nabla_\theta f(\hat{\theta}^t) = \hat{\theta}^t - \theta_* - \eta_t \mathbf{Q}^t \int_0^1 \nabla_\theta^2 f(\theta_* + \tau(\hat{\theta}^t - \theta_*))(\hat{\theta}^t - \theta_*)\, d\tau,$$

$$= \left( I - \eta_t \mathbf{Q}^t \int_0^1 \nabla_\theta^2 f(\theta_* + \tau(\hat{\theta}^t - \theta_*)) d\tau \right) (\hat{\theta}^t - \theta_*).$$

Since the projection $\mathcal{P}_\mathcal{C}$ in step 2 of the NewSamp can only decrease the $\ell_2$ distance, we obtain

$$\|\hat{\theta}^{t+1} - \theta_*\|_2 \leq \left\| I - \eta_t \mathbf{Q}^t \int_0^1 \nabla_\theta^2 f(\theta_* + \tau(\hat{\theta}^t - \theta_*)) d\tau \right\|_2 \|\hat{\theta}^t - \theta_*\|_2.$$

Note that the first term on the right hand side governs the convergence behavior of the algorithm.

In the following, we will provide the proof for sampling with replacement. The proof for the sampling without replacement follows from similar steps. For the relevant matrix concentration bounds using sampling without replacement, see i.e. [GN10].

Next, for an index set $S \subset [n]$, define the matrix $\mathbf{H}_S(\theta)$ as

$$\mathbf{H}_S(\theta) = \frac{1}{|S|} \sum_{i \in S} \mathbf{H}_i(\theta)$$

where $|S|$ denotes the size of the set. Denote the integral in the governing coefficient by $\widetilde{\mathbf{H}}$, that is,

$$\widetilde{\mathbf{H}} = \int_0^1 \nabla_\theta^2 f(\theta_* + \tau(\hat{\theta}^t - \theta_*)) d\tau.$$

By the triangle inequality, this coefficient can be bounded as

$$\left\| I - \eta_t \mathbf{Q}^t \widetilde{\mathbf{H}} \right\|_2 \leq \left\| I - \eta_t \mathbf{Q}^t \mathbf{H}_S(\hat{\theta}^t) \right\|_2 \tag{B.1}$$

$$+ \eta_t \left\| \mathbf{Q}^t \right\|_2 \left\{ \left\| \mathbf{H}_S(\hat{\theta}^t) - \mathbf{H}_{[n]}(\hat{\theta}^t) \right\|_2 + \left\| \mathbf{H}_{[n]}(\hat{\theta}^t) - \widetilde{\mathbf{H}} \right\|_2 \right\}.$$

Eq.(B.1) holds, regardless of the choice of $\mathbf{Q}^t$.

Using any indexing over the elements of $S$, we denote the each element in $S$ by $s_i$, i.e.,

$$S = \{s_1, s_2, ..., s_{|S|}\}.$$

Next, for a given $\theta$, we define the centered Hessians, $\mathbf{W}_i(\theta)$ as

$$\mathbf{W}_i(\theta) = \frac{1}{|S|} \{\mathbf{H}_{s_i}(\theta) - \mathbb{E}[\mathbf{H}_{s_i}(\theta)]\},$$

where the $\mathbb{E}[\mathbf{H}_{s_i}(\theta)]$ is just the full Hessian at $\theta$.

By the Assumption 2, we have

$$\max_{i \leq n} \|H_i(\theta)\|_2 = \left\|\boldsymbol{\nabla}_\theta^2 f_i(\theta)\right\|_2 \leq K, \tag{B.2}$$

$$\max_{i \leq n} \|\mathbf{W}_i\|_2 \leq \frac{2K}{|S|}, \qquad \max_{i \leq n} \left\|\mathbf{W}_i^2\right\|_2 \leq \frac{4K^2}{|S|^2}.$$

We apply the matrix Hoeffding's inequality [Tro12] and obtain for $\theta \in \mathcal{C}$,

$$\mathbb{P}\left(\left\|\mathbf{H}_S(\theta) - \mathbf{H}_{[n]}(\theta)\right\|_2 > \epsilon\right) \leq 2p \exp\left\{-\frac{\epsilon^2 |S|}{32 K^2}\right\}. \tag{B.3}$$

Therefore, to obtain a convergence rate of $\mathcal{O}(1/p)$, we let

$$\epsilon = C\sqrt{\frac{\log(p)}{|S|}}.$$

where $C = 8K$.

For the last term, we may write,

$$\begin{aligned}
\left\|\mathbf{H}_{[n]}(\hat{\theta}^t) - \widetilde{\mathbf{H}}\right\|_2 &= \left\|\mathbf{H}_{[n]}(\hat{\theta}^t) - \int_0^1 \boldsymbol{\nabla}_\theta^2 f(\theta_* + \tau(\hat{\theta}^t - \theta_*))d\tau\right\|_2, \\
&\leq \int_0^1 \left\|\mathbf{H}_{[n]}(\hat{\theta}^t) - \boldsymbol{\nabla}_\theta^2 f(\theta_* + \tau(\hat{\theta}^t - \theta_*))\right\|_2 d\tau, \\
&\leq \int_0^1 M_n(1-\tau)\|\hat{\theta}^t - \theta_*\|_2 d\tau, \\
&= \frac{M_n}{2}\|\hat{\theta}^t - \theta_*\|_2.
\end{aligned}$$

First inequality follows from the fact that norm of an integral is less than or equal to the integral of the norm. Second inequality follows from the Lipschitz property.

Combining the above results, we obtain the following for the governing term in Eq. (B.1): For some absolute constants $c, C > 0$, with probability at least $1 - 2/p$, we have

$$\left\|I - \eta_t \mathbf{Q}^t \mathbf{H}_{[n]}(\tilde{\theta}^t)\right\|_2 \leq \left\|I - \eta_t \mathbf{Q}^t \mathbf{H}_S(\hat{\theta}^t)\right\|_2 + \eta_t \left\|\mathbf{Q}^t\right\|_2 \left\{8K\sqrt{\frac{\log(p)}{|S|}} + \frac{M_n}{2}\|\hat{\theta}^t - \theta_*\|_2\right\}.$$

Hence, the proof is completed. $\qquad \square$

*Proof of Theorem 3.2.* Using the definition of $\mathbf{Q}^t$ in NewSamp, we immediately obtain that

$$\left\|I - \eta_t \mathbf{Q}^t \mathbf{H}_{S_t}(\hat{\theta}^t)\right\|_2 = \max_{i > r}\left\{\left|1 - \eta_t \frac{\lambda_i^t}{\lambda_{r+1}^t}\right|\right\}, \tag{B.4}$$

and that $\|\mathbf{Q}^t\|_2 = 1/\lambda_{r+1}^t$. Then the proof follows from Lemma 3.1 and by the assumption on the step size. $\qquad \square$

**Lemma B.1.** *Assume that the parameter set $\mathcal{C}$ is bounded, convex and $S_t \subset [n]$ is based on sub-sampling scheme S2. Further, let the Assumptions 1, 2 and 3 hold, almost surely. Conditioned on the event $\mathcal{E} = \{\theta_* \in \mathcal{C}\}$, for some absolute constants $c, c' > 0$ and $c_{\mathcal{E}} = c/\mathbb{P}(\mathcal{E})$, with probability at least $1 - c_{\mathcal{E}}\, e^{-p}$, updates of the form Eq. 1.2 satisfy*

$$\|\hat{\theta}^{t+1} - \theta_*\|_2 \leq \xi_1^t \|\hat{\theta}^t - \theta_*\|_2 + \xi_2^t \|\hat{\theta}^t - \theta_*\|_2^2,$$

*for coefficients $\xi_1^t, \xi_2^t$ defined as*

$$\xi_1^t = \left\| I - \eta_t \mathbf{Q}^t \mathbf{H}_{S_t}(\hat{\theta}^t) \right\|_2 + \eta_t \left\| \mathbf{Q}^t \right\|_2 \times c'K \sqrt{\frac{p}{|S_t|} \log\left( \frac{\mathrm{diam}(\mathcal{C})^2 \left(M_n + M_{|S_t|}\right)^2 |S_t|}{K^2} \right)},$$

$$\xi_2^t = \eta_t \frac{M_n}{2} \left\| \mathbf{Q}^t \right\|_2.$$

*Proof of Lemma B.1.* The first part of the proof is the same as Lemma 3.1 almost surely on the set $\mathcal{E}$. We carry our analysis from Eq. (B.1). Note that in this general set-up, the iterates are random variables that depend on the random functions. Therefore, we use a uniform bound for the right hand side in Eq.(B.1). That is, on $\mathcal{E}$

$$\left\| I - \eta_t \mathbf{Q}^t \mathbf{H}_{[n]}(\tilde{\theta}^t) \right\|_2 \leq \left\| I - \eta_t \mathbf{Q}^t \mathbf{H}_S(\hat{\theta}^t) \right\|_2$$

$$+ \eta_t \left\| \mathbf{Q}^t \right\|_2 \left\{ \sup_{\theta \in \mathcal{C}} \left\| \mathbf{H}_S(\theta) - \mathbf{H}_{[n]}(\theta) \right\|_2 + \frac{M_n}{2} \|\hat{\theta}^t - \theta_*\|_2 \right\}.$$

By the Assumption 1, given $\theta, \theta' \in \mathcal{C}$ such that $\|\theta - \theta'\|_2 \leq \Delta$, we have,

$$\left\| \mathbf{H}_S(\theta) - \mathbf{H}_{[n]}(\theta) \right\|_2 \leq \left\| \mathbf{H}_S(\theta') - \mathbf{H}_{[n]}(\theta') \right\|_2 + \left(M_n + M_{|S|}\right) \|\theta - \theta'\|_2$$

$$\leq \left\| \mathbf{H}_S(\theta') - \mathbf{H}_{[n]}(\theta') \right\|_2 + \left(M_n + M_{|S|}\right) \Delta.$$

Next, we will use a covering net argument to obtain a bound on the empirical process. Note that similar bounds on the matrix forms can be obtained through other approaches like *chaining* [DE15]. Let $\mathcal{T}_\Delta$ be a $\Delta$-net over the convex set $\mathcal{C}$. By the above inequality, we obtain

$$\sup_{\theta \in \mathcal{C}} \left\| \mathbf{H}_S(\theta) - \mathbf{H}_{[n]}(\theta) \right\|_2 \leq \max_{\theta' \in \mathcal{T}_\Delta} \left\| \mathbf{H}_S(\theta') - \mathbf{H}_{[n]}(\theta') \right\|_2 + \left(M_n + M_{|S|}\right) \Delta. \qquad \text{(B.5)}$$

In the following, we will argue that the right hand side is small with high probability using matrix concentration from [Tro12]. By the union bound over $\mathcal{T}_\Delta$, we have

$$\mathbb{P}\left( \max_{\theta' \in \mathcal{T}_\Delta} \left\| \mathbf{H}_S(\theta') - \mathbf{H}_{[n]}(\theta') \right\|_2 > \epsilon \right) \leq |\mathcal{T}_\Delta|\, \mathbb{P}\left( \left\| \mathbf{H}_S(\theta') - \mathbf{H}_{[n]}(\theta') \right\|_2 > \epsilon \right).$$

For the first term on the right hand side, by Lemma D.2, we write:

$$|\mathcal{T}_\Delta| \leq \left( \frac{\mathrm{diam}(\mathcal{C})}{2\Delta/\sqrt{p}} \right)^p.$$

As before, let $S = \{s_1, s_2, ..., s_{|S|}\}$, that is, $s_i$ denote the different indices in $S$. For any $\theta \in \mathcal{C}$ and $i = 1, 2, ..., n$, we define the centered Hessians $\mathbf{W}_i(\theta)$ as

$$\mathbf{W}_i(\theta) = \frac{1}{|S|} \left\{ \mathbf{H}_{s_i}(\theta) - \mathbf{H}_{[n]}(\theta) \right\}.$$

By the Assumption 2, we have the same bounds as in Eq. (B.2). Hence, for $\epsilon > 0$ and $\theta \in \mathcal{C}$, by the matrix Hoeffding's inequality

$$\mathbb{P}\left(\left\|\mathbf{H}_S(\theta) - \mathbf{H}_{[n]}(\theta)\right\|_2 > \epsilon\right) \leq 2p \exp\left\{-\frac{|S|\epsilon^2}{32K^2}\right\}.$$

We would like to obtain an exponential decay with a rate of at least $\mathcal{O}(p)$. Hence, we require,

$$p \log\left(\frac{\operatorname{diam}(\mathcal{C})\sqrt{p}}{2\Delta}\right) + \log(2p) + p \leq p \log\left(\frac{4\operatorname{diam}(\mathcal{C})\sqrt{p}}{\Delta}\right),$$

$$\leq \frac{|S|\epsilon^2}{32K^2},$$

which gives the optimal value of $\epsilon$ as

$$\epsilon \geq \sqrt{\frac{32K^2 p}{|S|} \log\left(\frac{4\operatorname{diam}(\mathcal{C})\sqrt{p}}{\Delta}\right)}.$$

Therefore, we conclude that for the above choice of $\epsilon$, with probability at least $1 - e^{-p}$, we have

$$\max_{\theta \in \mathcal{T}_\Delta} \left\|\mathbf{H}_S(\theta) - \mathbf{H}_{[n]}(\theta)\right\|_2 < \sqrt{\frac{32K^2 p}{|S|} \log\left(\frac{4\operatorname{diam}(\mathcal{C})\sqrt{p}}{\Delta}\right)}.$$

Applying this result to the inequality in Eq.(B.5), we obtain that with probability at least $1 - e^{-p}$,

$$\sup_{\theta \in \mathcal{C}} \left\|\mathbf{H}_S(\theta) - \mathbf{H}_{[n]}(\theta)\right\|_2 \leq \sqrt{\frac{32K^2 p}{|S|} \log\left(\frac{4\operatorname{diam}(\mathcal{C})\sqrt{p}}{\Delta}\right)} + \left(M_n + M_{|S|}\right)\Delta.$$

The right hand side of the above inequality depends on the net covering diameter $\Delta$. We optimize over $\Delta$ using Lemma D.3 which provides for

$$\Delta = 4\sqrt{\frac{K^2 p}{\left(M_n + M_{|S|}\right)^2 |S|} \log\left(\frac{\operatorname{diam}(\mathcal{C})^2 \left(M_n + M_{|S|}\right)^2 |S|}{K^2}\right)},$$

we obtain that with probability at least $1 - e^{-p}$,

$$\sup_{\theta \in \mathcal{C}} \left\|\mathbf{H}_S(\theta) - \mathbf{H}_{[n]}(\theta)\right\|_2 \leq 8K\sqrt{\frac{p}{|S|} \log\left(\frac{\operatorname{diam}(\mathcal{C})^2 \left(M_n + M_{|S|}\right)^2 |S|}{K^2}\right)}.$$

Combining this with the bound stated in Eq.(B.1), and taking the conditioning on the set $\mathcal{E}$ into account, we conclude the proof. $\qquad\square$

*Proof of Theorem 3.4.*

$$\left|\xi_1^t - \xi_1^*\right| = \left|\frac{\lambda_p^t}{\lambda_{r+1}^t} - \frac{\lambda_p^*}{\lambda_{r+1}^*}\right| + cK\sqrt{\frac{\log(p)}{|S_t|}} \left|\frac{1}{\lambda_{r+1}^t} - \frac{1}{\lambda_{r+1}^*}\right|$$

$$\leq \frac{K|\lambda_{r+1}^t - \lambda_{r+1}^*| + K|\lambda_p^t - \lambda_p^*|}{\lambda_{r+1}^* \lambda_{r+1}^t} + cK\sqrt{\frac{\log(p)}{|S_t|}} \frac{|\lambda_{r+1}^t - \lambda_{r+1}^*|}{\lambda_{r+1}^* \lambda_{r+1}^t}$$

Figure 3: The plots demonstrate the behavior of several optimization methods on a synthetic data set for training SVMs. The elapsed time in seconds versus log of $\ell_2$-distance to the true minimizer is plotted. Red color represents the proposed method NewSamp .

By the Weyl's and matrix Hoeffding's [Tro12] inequalities (See Eq. (B.3) for details), we can write

$$
|\lambda_j^t - \lambda_j^*| \leq \left\| \mathbf{H}_{S_t}(\hat{\theta}^t) - \mathbf{H}_{[n]}(\theta_*) \right\|_2 \leq cK\sqrt{\frac{\log(p)}{|S_t|}},
$$

with probability $1 - 2/p$. Then,

$$
\begin{aligned}
\left| \xi_1^t - \xi_1^* \right| &\leq \frac{c'K\sqrt{\frac{\log(p)}{|S_t|}}}{\lambda_{r+1}^* \lambda_{r+1}^t} + \frac{c''K^2 \frac{\log(p)}{|S_t|}}{\lambda_{r+1}^* \lambda_{r+1}^t}, \\
&\leq \frac{c'''K\sqrt{\frac{\log(p)}{|S_t|}}}{k\left(k - cK\sqrt{\frac{\log(p)}{|S_t|}}\right)},
\end{aligned}
$$

for some constants $c$ and $c'''$. $\qquad\square$

*Proof of Corollary 4.1.* Observe that $f_i(\theta) = \Phi(\langle x_i, \theta \rangle) - y_i \langle x_i, \theta \rangle$, and $\nabla_\theta^2 f_i(\theta) = x_i x_i^T \Phi^{(2)}(\langle x_i, \theta \rangle)$. For an index set $S$, we have $\forall \theta, \theta' \in \mathcal{C}$

$$
\begin{aligned}
\|\mathbf{H}_S(\theta) - \mathbf{H}_S(\theta')\|_2 &= \left\| \frac{1}{|S|} \sum_{i \in S} x_i x_i^T \left[ \Phi^{(2)}(\langle x_i, \theta \rangle) - \Phi^{(2)}(\langle x_i, \theta' \rangle) \right] \right\|_2, \\
&\leq L \max_{i \in S} \|x_i\|_2^3 \|\theta - \theta'\|_2 \leq L R_x^{3/2} \|\theta - \theta'\|_2.
\end{aligned}
$$

Therefore, the Assumption 1 is satisfied with the Lipschitz constant $M_{|S_t|} := L R_x^{3/2}$. Moreover, by the inequality

$$
\left\| \nabla_\theta^2 f_i(\theta) \right\|_2 = \|x_i\|_2^2 \, \Phi^{(2)}(\langle x_i, \theta \rangle) \leq R_x, = \left\| x_i x_i^T \Phi^{(2)}(\langle x_i, \theta \rangle) \right\|_2
$$

the Assumption 2 is satisfied for $K := R_x$. We conclude the proof by applying Theorem 3.2. $\qquad\square$

## C  Further experiments and details

In this section, we present the details of the experiments presented in Figure 2 and provide additional simulation results. In the experiments, R is used as the programming platform.

Logistic Regression

| | Rank=3 | | Rank=10 | | Rank=20 | |
|---|---|---|---|---|---|---|
| Method | Elapsed(sec) | Iter | Elapsed(sec) | Iter | Elapsed(sec) | Iter |
| NewSamp | 26.412 | 12 | 32.059 | 15 | 55.995 | 26 |
| BFGS | 50.699 | 22 | 54.756 | 31 | 56.606 | 34 |
| LBFGS | 103.590 | 47 | 64.617 | 37 | 107.708 | 67 |
| Newton | 18235.842 | 449 | 35533.516 | 941 | 31032.893 | 777 |
| GD | 345.025 | 198 | 322.671 | 198 | 311.946 | 197 |
| AGD | 449.724 | 233 | 436.282 | 272 | 450.734 | 290 |

Support Vector Machines

| | Rank=3 | | Rank=10 | | Rank=20 | |
|---|---|---|---|---|---|---|
| Method | Elapsed(sec) | Iter | Elapsed(sec) | Iter | Elapsed(sec) | Iter |
| NewSamp | 47.755 | 8 | 52.767 | 9 | 124.989 | 22 |
| BFGS | 13352.254 | 2439 | 10672.657 | 2219 | 21874.637 | 4290 |
| LBFGS | 326.526 | 67 | 218.706 | 44 | 275.991 | 55 |
| Newton | 775.191 | 16 | 734.480 | 16 | 4159.486 | 106 |
| GD | 1512.305 | 238 | 1089.413 | 237 | 1518.063 | 269 |
| AGD | 1695.44 | 239 | 1066.484 | 238 | 1874.75 | 294 |

Table 2: Details of the simulations presented in Figures 3.

We first start with additional experiments. The goal of this experiment is to further analyze the effect of rank in the performance of NewSamp . We experimented using $r$-spiked model for $r = 3, 10, 20$. The case $r = 3$ was already presented in Figure 2, which is included in Figure 3 to ease the comparison. The results are presented in Figures 3 and the details are summarized in Table 2. In the case of LR optimization, we observe through Figure 3 that stochastic algorithms enjoy fast convergence in the beginning but slows down later as they get close to the true minimizer. The algorithms that come closer to NewSamp in terms of performance are BFGS and LBFGS. Especially when $r = 20$, performance of BFGS and that of NewSamp are similar, yet NewSamp still does better. In the case of SVM optimization, the algorithm that comes closer to NewSamp is Newton's method.

Further, we demonstrate how the convergence coefficients $\xi_1$ and $\xi_2$ vary over several datasets in Figure 4.

## D    Useful lemmas

**Lemma D.1.** *Let $X$ be a symmetric $p \times p$ matrix, and let $\mathcal{T}_\epsilon$ be an $\epsilon$-net over $S^{p-1}$. Then,*

$$\|X\|_2 \leq \frac{1}{1 - 2\epsilon} \sup_{v \in \mathcal{T}_\epsilon} |\langle Xv, v \rangle| .$$

**Lemma D.2.** *Let $\mathcal{C}$ be convex and bounded set in $\mathbb{R}^p$ and $T_\epsilon$ be an $\epsilon$-net over $\mathcal{C}$. Then,*

$$|T_\epsilon| \leq \left( \frac{\text{diam}(\mathcal{C})}{2\epsilon/\sqrt{p}} \right)^p .$$

*Proof of Lemma D.2.* The set $\mathcal{C}$ can be contained in a $p$-dimensional cube of size $\text{diam}(\mathcal{C})$. Consider a grid over this cube with mesh width $2\epsilon/\sqrt{p}$. Then $\mathcal{C}$ can be covered with at most $(\text{diam}(\mathcal{C})/(2\epsilon/\sqrt{p}))^p$ many cubes of edge length $2\epsilon/\sqrt{p}$. If ones takes the projection of the centers of such cubes onto $\mathcal{C}$ and considers the circumscribed balls of radius $\epsilon$, we may conclude that $\mathcal{C}$ can be covered with at most

$$\left( \frac{\text{diam}(\mathcal{C})}{2\epsilon/\sqrt{p}} \right)^p$$

many balls of radius $\epsilon$. □

**Lemma D.3.** *For $a, b > 0$, and $\epsilon$ satisfying*

$$\epsilon = \left\{ \frac{a}{2} \log \left( \frac{2b^2}{a} \right) \right\}^{1/2} \quad and \quad \frac{2}{a} b^2 > e,$$

CT Slices Dataset

| | LR | | SVM | |
|---|---|---|---|---|
| Method | Elapsed(sec) | Iter | Elapsed(sec) | Iter |
| NewSamp | 9.488 | 19 | 22.228 | 33 |
| BFGS | 9.568 | 38 | 2094.330 | 5668 |
| LBFGS | 51.919 | 217 | 165.261 | 467 |
| Newton | 14.162 | 5 | 58.562 | 25 |
| GD | 350.863 | 2317 | 1660.190 | 4828 |
| AGD | 176.302 | 915 | 1221.392 | 3635 |

MSD Dataset

| | LR | | SVM | |
|---|---|---|---|---|
| Method | Elapsed(sec) | Iter | Elapsed(sec) | Iter |
| NewSamp | 25.770 | 38 | 71.755 | 49 |
| BFGS | 43.537 | 75 | 9063.971 | 6317 |
| LBFGS | 81.835 | 143 | 429.957 | 301 |
| Newton | 144.121 | 30 | 100.375 | 18 |
| GD | 642.523 | 1129 | 2875.719 | 1847 |
| AGD | 397.912 | 701 | 1327.913 | 876 |

Synthetic Dataset

| | LR | | SVM | |
|---|---|---|---|---|
| Method | Elapsed(sec) | Iter | Elapsed(sec) | Iter |
| NewSamp | 26.412 | 12 | 47.755 | 8 |
| BFGS | 50.699 | 22 | 13352.254 | 2439 |
| LBFGS | 103.590 | 47 | 326.526 | 67 |
| Newton | 18235.842 | 449 | 775.191 | 16 |
| GD | 345.025 | 198 | 1512.305 | 238 |
| AGD | 449.724 | 233 | 1695.44 | 239 |

Table 3: Details of the experiments presented in Figure 2.

Figure 4: The plots demonstrate the behavior of $\xi_1$ and $\xi_2$ over several datasets.

*we have $\epsilon^2 \geq a \log(b/\epsilon)$.*

*Proof.* Since $a, b > 0$ and $x \rightarrow e^x$ is a monotone increasing function, the above inequality condition is equivalent to

$$\frac{2\epsilon^2}{a} e^{\frac{2\epsilon^2}{a}} \geq \frac{2b^2}{a}.$$

Now, we define the function $f(w) = we^w$ for $w > 0$. $f$ is continuous and invertible on $[0, \infty)$. Note that $f^{-1}$ is also a continuous and increasing function for $w > 0$. Therefore, we have

$$\epsilon^2 \geq \frac{a}{2} f^{-1}\left(\frac{2b^2}{a}\right)$$

Observe that the smallest possible value for $\epsilon$ would be simply the square root of $af^{-1}\left(2b^2/a\right)/2$. For simplicity, we will obtain a more interpretable expression for $\epsilon$. By the definition of $f^{-1}$, we have

$$\log(f^{-1}(y)) + f^{-1}(y) = \log(y).$$

Since the condition on $a$ and $b$ enforces $f^{-1}(y)$ to be larger than 1, we obtain the simple inequality that

$$f^{-1}(y) \leq \log(y).$$

Using the above inequality, if $\epsilon$ satisfies

$$\epsilon^2 = \frac{a}{2} \log\left(\frac{2b^2}{a}\right) \geq \frac{a}{2} g\left(\frac{2b^2}{a}\right),$$

we obtain the desired inequality. $\qquad\square$