[Reviews · NeurIPS 2015]

Submitted by Assigned_Reviewer_1

A hat is missing in (1.2)

Quadratic at start is a bad feature because this means that one should provide a good initial guess as shown in Corollary 3.7. The authors should not present it as a good feature.
Summary: This paper presents a quasi-Newton method where the Hessian matrix is estimated by sub-sampling. This sub-sampled matrix is processed to improved its spectrum properties and thus lead to an efficient algorithm, at least in the case when n is much larger than p. The authors present well the limits and advantages of the method and give convincing numerical experiments.

Submitted by Assigned_Reviewer_2

The paper develops a Newton method based on sub-sampling and applying a low-rank approximation to the Hessian and establishes its convergence rate.

Prior to this work, there does not seem to have any convergence rate analysis of sub-sampling-based Newton methods.

The current paper makes an important contribution in this regard.

Moreover, extensive numerical experiments verify the efficiency of the proposed method.
Summary: The results in the paper are strong and the techniques will find further applications in the development of fast Newton-type methods for big data applications.

Submitted by Assigned_Reviewer_3

Newton-type method are becoming more and more popular even for large-scale problems. The main challenge is to obtain a better descent direction than the negative gradient direction by using information from the second-order derivatives. This is achieved in the algorithm NewSamp proposed in this paper by inverting a subsampled and regularized version of the Hessian. This idea have been used before but this paper contributes an interesting convergence analysis. It is shown that, first, the algorithm inherits the quadratic convergence of Newton's method which decreases to a linear convergence rate as the number of iteration increases.

The paper analyses this in a nice way: Different sampling techniques, The convergence rate coefficients are given explicitly and it is shown how to obtain reasonable parameters.

When compared to the standard (projected) Newton method, NewSamp is different in two aspects:

* In each step the Hessian is approximated from a minibatch (subsampling)

* A "regularized" version of the subsampled Hessian is inverted (Isn't P_r(H_{S_t} - \lambda_{r+1}I) = P_r(H_{S_t})?).

This is presented nicely in the paper. However, the comparison to more similar techniques would be worthwhile. For example, the algorithm in [Mar10] also subsamples the Hessian and uses regularization ("damping"). However, in [Mar10] the approximation of the Hessian is not inverted exactly but approximated since only its application to the gradient is needed. It remains unclear whether this could give better results even for the relatively small values of p considered in this paper and whether this additional inexactness influences the convergence rate results. Similarly, [BHNS14] might perform better than the BFGS algorithm used in the experiments.

The presentation of the experimental results is a bit confusing: It is unclear how exactly the synthetic data was produced and whether the time-vs.-error plots were obtained by averaging over several runs. Concerning the sample size of NewSamp used in the experiments, one is referred to 3.4 but there it is only given as an order of magnitude. It is a bit unclear what "r" and "rank" refer to. If it refers to the rank of the low-rank approximation of the sampled Hessian in NewSamp then it us unclear why all algorithms are dependent on it (Figure 3, Table 2).

Note that there is probably a typo in the second table of Table 2: For Rank=10 the NewSamp and BFGS results were mixed up.
Summary: This paper deals with the application of Newton's method to the minimization of functions which depends on a large number of data points of moderate dimension. In particular, the paper shows the convergence behavior when the Hessian matrix is approximated from a limited number of data points in each iteration.

Submitted by Assigned_Reviewer_4

This paper analyzes the convergence behavior of a sub-sampled Newton method. Simulation results show the high-performance of the sub-sampled Newton method in compare to well-known optimization procedures.

The major contribution of the paper is its convergence analysis. I guess, however, that there is a flaw in the convergence analysis. In the line 487 of appendix, the authors invoke a mean value theorem. To the best of my knowledge, this integral version of the mean value theorem does not hold for vector-valued (matrix-valued) functions. This flaw affects the subsequent analysis and questions the main contribution of the paper.

The paper is well-written and self-contained. I did not see any typographical and grammatical problems in the paper. The authors, however, change the width of the paper so their material fit to the 8-pages limit. They should use the nips style and subsequently shorten the paper or put some more materials into the appendix so that the paper fits the nips style with 8 pages limit. The authors also changed the numerical citation style of nips and this should be corrected too.

After Authors Rebuttal: The authors fixed this problem with their proof in the rebuttal.
Summary: The simulation results show the interesting behavior of the sub-sampled Newton method. If the authors could solve the problems with the convergence analysis, this paper could make an interesting contribution to the machine-learning and optimization communities.

Author Feedback
Author rebuttal: We thank the reviewers for their thoughtful comments. Please find our detailed response below.

**Assigned_Reviewer_2:
-A "regularized" version:
Please note that the largest eigenvalue of P_r(H_{S_t} - \lambda_{r+1}I) is just the largest eigenvalue of P_r(H_{S_t}) minus \lambda_{r+1}.

-Further comparisons:
We also think that the Experiments section would benefit from additional comparisons with algorithms exploiting conjugate gradient(CG) methods as in [Mar10]. The performance of CG based algorithms rely on the approximation obtained by the CG method, which in turn depends on the underlying spectrum of the Hessian. Convergence rates of CG methods are known to be fast if the spectrum of the Hessian is spread but clustered. We will include one of CG based methods into our experiments along with a comparison with the stochastic BFGS [BHNS14].

-Synthetic data:
Data is generated through a multivariate normal distribution with mean 0 and the covariance matrix \Sigma follows a spiked model, i.e. \Sigma has a few strong eigenvalues and rest of them are small and can be considered as noise [DGJ13]. We discussed this in Appendix but it needs to be clarified in the main text as well.

-Time vs error plots:
Plots are obtained by a single run. This is consistent with the real data experiments.

-Sample size:
In line 214, if we consider the term involving |S_t|, we observe that |S_t| must satisfy |S_t| > c^2 (K/\lambda_{r+1})^2 \ log(p). This is the suggested sample size in Section 3.4. In theory, c=8 is sufficient in the above formulation. In the experiments, we used c=3 to choose the sample size.

-Rank in plots:
'Rank' refers to the rank threshold (also denoted as r) which is an input to the NewSamp algorithm (line 104). This parameter can be considered as the number of "strong" eigenvalues in the Hessian. As it depends on the spectral futures of the Hessian, it should be specific to each dataset. Section 3.4 provides a discussion on how to choose this parameter.

We will clarify these points in an updated version of the paper.

**Assigned_Reviewer_3:
We thank the reviewer for bringing this flaw to our attention. We can
fix this issue by adding a line to the proof.
[Note that rest of the proof and the statement of the theorem remain unchanged]
The correction is as follows:

1- Eliminate line 487.
2- Use triangle inequality in line 491 with H_{[n]}(\tilde{\theta}) replaced by the integral in line 487.
3- After triangle inequality, the term involving the integral can be
upper bounded by moving the integration out of the norms (norm of an
integral is at most the integral of a norm).
4- Result will follow by exploiting either Lipschitz property or the mean
value theorem (applied to the integral of the norm, that is a scalar).

We also provide the LaTeX write up for convenience.

%
% \usepackage{amsmath} % for align
\begin{align}
&\left\| I - \eta_tQ^t
\int_0^1\nabla_\theta f (\theta_* + \tau(\hat{\theta}^{t}-\theta_*))d\tau\right\|_2
\leq \left\| I - \eta_tQ^tH_S(\hat{\theta}^t)\right \|_2 \\
&+ \eta_t\left\|Q^t\right\|_2
\Big\{ \left\|H_S(\hat{\theta}^t)- H_{[n]}(\hat{\theta}^t) \right\|_2
+ \left\|H_{[n]}(\hat{\theta}^t) -
\int_0^1\nabla^2_\theta f (\theta_* + \tau(\hat{\theta}^{t}-\theta_*))d\tau
\right\|_2 \Big\}
\end{align}
After moving the integration out of norm and using the Lipschitz property, we obtain
\begin{align}
\left\|H_{[n]}(\hat{\theta}^t) -
\int_0^1\nabla^2_\theta f (\theta_* + \tau(\hat{\theta}^{t}-\theta_*))d\tau
\right\|_2 \leq
\int_0^1\left\|H_{[n]}(\hat{\theta}^t) -
\nabla^2_\theta f (\theta_* + \tau(\hat{\theta}^{t}-\theta_*))
\right\|_2 d\tau \\
\leq \int_0^1 M_n (1-\tau)\|\hat{\theta}^t - \theta_* \|_2 d\tau \\
= M_n \|\hat{\theta}^t - \theta_* \|_2 /2
\end{align}
%

**Assigned_Reviewer_6:
Similar to Newton's method, NewSamp's quadratic convergence is a local property. As the reviewer points out, quadratic convergence requires good initial guess. If the initial point is not close to the optimum, NewSamp (or Newton's method) will start with a "damped" phase (provided with a suitable step size). In this case, the convergence phases of NewSamp will be:
damped -> quadratic -> linear.
[Boyd, Vandenberghe 04 - p488] explains this phenomenon in detail for the Newton's method, but the idea works for most second order methods.

We will address the reviewer's concern and discuss this in detail.